# Comparison between Intravoxel Incoherent Motion and Splenic Volumetry to Predict Hepatic Fibrosis Staging in Preoperative Patients

**DOI:** 10.3390/diagnostics13203200

**Published:** 2023-10-13

**Authors:** Takayuki Arakane, Masahiro Okada, Yujiro Nakazawa, Kenichiro Tago, Hiroki Yoshikawa, Mariko Mizuno, Hayato Abe, Tokio Higaki, Yukiyasu Okamura, Tadatoshi Takayama

**Affiliations:** 1Department of Radiology, Nihon University School of Medicine, Tokyo 173-8610, Japan; arakane.takayuki@nihon-u.ac.jp (T.A.);; 2Department of Digestive Surgery, Nihon University School of Medicine, Tokyo 173-8610, Japan

**Keywords:** hepatic fibrosis, diffusion-weighted imaging, intravoxel incoherent motion, magnetic resonance imaging (MRI), computed tomography (CT)

## Abstract

Intravoxel incoherent motion (IVIM) and splenic volumetry (SV) for hepatic fibrosis (HF) prediction have been reported to be effective. Our purpose is to compare the HF prediction of IVIM and SV in 67 patients with pathologically staged HF. SV was divided by body surface area (BSA). IVIM indices, such as slow diffusion-coefficient related to molecular diffusion (D), fast diffusion-coefficient related to perfusion in microvessels (D*), apparent diffusion-coefficient (ADC), and perfusion related diffusion-fraction (f), were calculated by two observers (R1/R2). D (*p* = 0.718 for R1, *p* = 0.087 for R2) and D* (*p* = 0.513, *p* = 0.708, respectively) showed a poor correlation with HF. ADC (*p* = 0.034, *p* = 0.528, respectively) and f (*p <* 0.001, *p* = 0.007, respectively) decreased as HF progressed, whereas SV/BSA increased (*p* = 0.015 for R1). The AUCs of SV/BSA (0.649–0.698 for R1) were higher than those of f (0.575–0.683 for R1 + R2) for severe HF (≥F3–4 and ≥F4), although AUCs of f (0.705–0.790 for R1 + R2) were higher than those of SV/BSA (0.628 for R1) for mild or no HF (≤F0–1). No significant differences to identify HF were observed between IVIM and SV/BSA. SV/BSA allows a higher estimation for evaluating severe HF than IVIM. IVIM is more suitable than SV/BSA for the assessment of mild or no HF.

## 1. Introduction

Hepatic fibrosis (HF) is an important factor in patients with chronic liver disease and those requiring a surgical operation of the liver [1] because HF may lead to surgical restrictions and affect the patient’s prognosis. A liver biopsy for histopathological assessment is commonly performed to stage HF. However, a biopsy can cause complications such as haemorrhage and infection, as well as inherent drawbacks, including sampling error. Therefore, non-invasive imaging-based methods [2,3,4] have been developed to assess HF. Diffusion-weighted imaging (DWI) represents the motion or diffusivity of the molecules. The apparent diffusion-coefficient (ADC) is calculated as the diffusivity and consists of substantial molecular diffusion in solid tissues and molecular movements in the vascular microcirculation (perfusion) [5]. Intravoxel incoherent motion (IVIM) is used for microscopic translations which occur in voxels on MRI [6], and the restricted diffusion observed in patients with cirrhosis may be related to D* variations according to Luciani et al. [7] and Ichikawa et al. [4]. Several researchers have reported that the ADC value is significantly reduced compared to the non-cirrhotic liver [8,9,10]. The measurement of splenic volumetry (SV) is simple, SV/body surface area (BSA) is a better predictor of HF, and SV is related to the severity of HF [11,12]. Although analyses by IVIM and SV are promising techniques used for staging HF, to our knowledge, no comparative study of these analyses has been conducted. This study aimed to compare IVIM imaging and SV measurements for staging HF.

## 2. Materials and Methods

### 2.1. Patients

This retrospective study was performed in accordance with the principles of the Declaration of Helsinki, and approved by the relevant Institutional Review Board of Nihon University Itabashi Hospital (RK-210413-9). Written informed consent was waived by the Institutional Review Board (Nihon University Itabashi Hospital, Clinical Research Judging Committee) because of the retrospective nature of this study. However, informed consent was obtained for MR and CT examinations and for publication of the patient’s images. This study included 88 consecutive patients who underwent liver MRI including 10-b-values DWI from November 2018 to January 2020. The inclusion criteria for our study were as follows: consecutive patients aged ≥ 20 years who were candidates for initial liver resection for liver tumour and those with available IVIM and dynamic CT imaging data within 3 months prior to liver resection. The exclusion criteria were as follows: patients without liver resection because of clinical or biochemical evidence of decompensated liver function (Child–Pugh classification C, ICG-R15 ≥ 35%, or serum total bilirubin level ≥ 2.0 mg/dL), tumour status, or portal hypertension (including the presence of high-risk oesophageal varices), and those who had already undergone hepatectomy or splenectomy prior to the IVIM-MRI and CT.

### 2.2. IVIM Imaging

Discovery 750 3.0T (GE Medical Systems, Chicago, IL, USA) with GEM Body Array (GEM Anterior Array + GEM Posterior Array), Ingenia 3.0T, and Achieva 1.5T Nova (Philips Healthcare, Best, The Netherlands) with SENSE Torso/Cardiac Coil were used for IVIM imaging, which was acquired in the transverse plane by respiratory-triggered fat-saturated spin echo-echo planar imaging. Motion-proving gradient pulses were applied concurrently in three directions (x, y, and z). The IVIM parameters are listed in Table 1 and Table 2.

The calculation for IVIM was performed using the Synapse Vincent software version 5.5 (Fujifilm Medical, Tokyo, Japan), which provided the D, D*, ADC, and f parameters mapped on a pixel-by-pixel basis (Figure 1).

The decrease in signal intensity of the hepatic lesions was compared with that of the spleen. We estimated signal attenuation using the following equation [6,13]:SI/SIo = exp (−b∙ADC)
SI/SIo = (1 − f) ∙ exp (−bD) + f ∙ exp (−bD*), as shown in a previous report,
where D and D* are the slow diffusion-coefficients related to molecular diffusion, and the fast diffusion-coefficient related to perfusion in micro-vessels, respectively, and f is the perfusion-related diffusion fraction.

To obtain IVIM parameters, three regions of interest (ROIs) were placed in the liver parenchyma (left lobe, anterior right lobe, and posterior right lobe), excluding artefacts, large intrahepatic vessels, and liver tumours, by two radiologists (T. A. and M. O.) with 5 years and 25 years of experience in abdominal radiology, who were blinded to the clinical, surgical, and pathological results. The measurements from three ROIs were averaged and used as the patient results. The region of interest (ROI) was drawn to prevent a peripheral liver zone of <1 cm. Each radiologist independently placed the ROIs. Circular ROIs with a size of 10 mm were placed at the hilum level of the liver. The measurements were recorded and compared separately.

### 2.3. CT Volumetry Analysis for Spleen

SV was measured using a viewer workstation (Synapse 3D^®^ ver5.5, Fujifilm Medical) to perform 3D reconstruction using dynamic liver CT data. An abdominal radiologist (T. A) with 5 years of experience made CTV images of the spleens. Without recognition of other organs in close proximity, SV measurements (Figure 2) were performed at approximately 1 min using the SAI viewer^®^ (Fujifilm Medical), which uses Deep Learning technology. The SV was corrected using BSA (TLV/BSA, SV/BSA) [14]. BSA was calculated using the Dubois formula (BSA [m^2^] = 0.007184 × height [cm] ^0.725^ × weight [kg] ^0.425^) [14].

## 3. Pathology

From surgically resected specimens for all patients, the pathological evaluation for HF was performed by two pathologists using the New Inuyama Classification [15]: F0, no fibrosis; F1, fibrous portal expansion; F2, bridging fibrosis; F3, bridging fibrosis with architectural distortion; F4, cirrhosis.

### Statistical Analysis

The intraclass-correlation-coefficients (ICCs) were used for interrater reliability, with an ICC of <0.50 defined as poor; 0.50–0.74 as moderate; 0.75–0.90 as good; and ≥0.90, excellent [16]. The mean and standard deviation of the IVIM and SV/BSA were calculated for each group. The correlation between the results of each imaging analysis and the HF stage was assessed using Spearman’s rank correlation. Receiver operating characteristic curve analysis was performed to determine the accuracy of IVIM and SV/BSA in staging HF. The highest AUCs for IVIM and SV/BSA were compared for HF grading using the Delong test. SPSS Version 27.0 (IBM Corp., Armonk, NY, USA) was used for analysis.

## 4. Results

### 4.1. Patients

This study included 88 patients who underwent hepatic surgery; however, 20 patients were excluded because the interval between IVIM and surgery was longer than the inclusion criteria or image deficiency obtained. One of the remaining 68 patients was excluded from the study because he had undergone a splenectomy. Finally, 67 patients were included in the study (Figure 3). The patient characteristics are shown in Table 3.

### 4.2. IVIM Analysis for Liver

The IVIM parameters, such as D, D*, ADC, and f, at each fibrosis grade determined by the two observers are summarised in Table 4. The hierarchisation for each HF stage was clear in the ADC for R1 and f for both R1 and R2. IVIM_f was significantly associated with the grading of HF by both observers (*p* < 0.001 and *p* = 0.007 by Spearman’s rank correlation coefficient test). ADC was significantly correlated with HF (*p* = 0.034) in only one observer. Other parameters of IVIM showed no significant differences in HF grading. Box-and-whisker diagrams for each parameter measurement are shown in Figure 4. When the degree of agreement between the two observers was evaluated using the intra-class correlation coefficient, poor reproducibility of IVIM measurements between the two observers was observed due to an ICC of less than 0.7, as shown in Table 5.

### 4.3. CT Volumetry Analysis for Spleen

SV/BSA increased with the exacerbation of HF. The results of the SV/BSA measurements at each fibrosis grade are shown in Table 4. SV/BSA was significantly associated with HF grading (*p* = 0.015, Spearman’s rank correlation coefficient test).

A box-and-whisker diagram for each parameter measurement is shown in Figure 4. The hierarchisation for each stage of HF was clear in the SV/BSA.

### 4.4. Comparison between IVIM and SV/BSA for Estimation of Liver Fibrosis and Cirrhosis

#### ROC Analysis

The ability of IVIM_f and SV/BSA to differentiate HF, which were both significantly correlated with the severity of HF in the assessment of the two observers in IVIM, was investigated using ROC analysis.

The comparisons between SV/BSA and f are shown in Table 6. There was a high ability to identify none or mild HF (≤F0–1) stage in the f of IVIM (AUC: 0.790 for R1, 0.625 for R2). However, there was also the ability to identify severe HF stages (≥F3–4, ≥F4) in SV/BSA (AUC: 0.698 for ≥F3–4, 0.649 for ≥F4). SV/BSA showed stable AUC values with a relatively high distinguishing ability (AUC; 0.628–0.698) for severe HF stages (≥F3–4), although the distinguishing ability was not high for none or mild HF (≤F0–1). The f of IVIM showed a high distinguishing ability (AUC; 0.705–0.790) for none or mild HF (≤F0–1), but a low distinguishing ability for severe HF stages (AUC; 0.575–0.620 for ≥F3–4) and unstable AUCs overall.

## 5. Discussion

Previous studies have shown that HF is associated with decreased D*, f, and ADC values [17,18]. According to Dyvorne et al. [19], the IVIM parameters f and D decrease as the HF progresses. In our results, f was especially useful for estimating HF, especially for no or mild HF. Previous reports have shown that D*, f, and ADC values were reduced as HF increased [7,17,18]. Thus, as to which parameter of IVIM best reflects HF remains controversial. Considering the ICC confidence intervals, the ICCs of ADC and f were moderate (below 0.50–0.74), whereas those of D and D* were poor (below 0.50). Manual registration of the ROI in IVIM is prone to interobserver variability and errors [20].

The AUC of SV/BSA was good (0.76–0.83) to estimate HF in a previous report [11], but the AUC was 0.628–0.698 in the present study was lower than that of the previous study. This may be attributed to the fact that portal hypertension was mild in most cases, and splenomegaly was sufficient to buffer portal blood flow because SV was associated with the degree of portal hypertension.

There were significant differences in patient background factors such as age, Plt, Alb, ALBI, APRI, and FIB-4. A previous report [11] showed lower SV and SV/BSA than the current report, although SV and SV/BSA did not show significant differences (Appendix A; this Table is a comparison using the original data from our previously published paper [11]). Categorical variables, such as sex, HBV, HCV, alcoholic liver disease, and Child–Pugh classification score, showed no significant differences between the previous [11] and the current study. However, the previous report showed significantly higher APRI and FIB-4 than the current report; thus, the previous study included patients with relatively poor liver function.

IVIM showed a large difference between the two measurements in two observers; the location of the ROI and different ROI sizes were likely to cause differences in measurements. This would result in a quantitative assessment with low reproducibility. CTVs, such as splenic volumetry, can be accurately assessed using artificial intelligence technology, the so-called SAI, because the imaging workstation enables semi-automatic and rapid reconstruction.

Spleen enlargement is caused by portal hypertension or cirrhosis [21]. We believe that SV/BSA is suitable for estimating HF and can contribute to safe operative management because highly fibrotic livers are known to be at risk for severe complications after surgery [22].

In particular, SV/BSA is a better predictor of HF than IVIM as an imaging parameter that can predict severe HF (≥F3). This result is in line with the previous considerations that SV/BSA is superior to extracellular volume fraction (ECV) in estimating HF [11]. The correction by BSA was used because SV may be affected by differences in patient body size, although SV is unrelated to patient height and weight [23]. SV/BSA was useful for predicting severe HF (≥F3), with an AUC of 0.698. This is consistent with the results of previous reports [24,25]. Without the need to use a liver biopsy or MR elastography, CT volume biomarkers can be obtained retrospectively with routine scans obtained for other indications [24,25]. SV can be adapted to patients with poor renal function because it can be measured even with CT without contrast media. However, it must be noted that there are other causes of large SV besides cirrhosis, such as haematological diseases and infections.

Non-invasive methods such as US elastography, MRI-elastography, and laboratory tests are used to evaluate HF. Ultrasonography (US) has non-invasive character, and FibroScan or Transient Elastography show an excellent ability of liver stiffness, such as the sensitivity of 96.2% and the specificity of 92.2% for fibrosis stage ≥ 4 [26]. MR elastography is a useful tool for assessing pathological conditions that affect the elasticity of soft tissues, such as HF [27]. While MR elastography is non-invasive and provides accurate staging of HF [28], its widespread adoption in all hospitals is hindered by the requirement for additional vibration equipment. In recent years, there have been efforts to estimate the degree of HF using existing imaging and blood tests, such as aspartate aminotransferase-platelet ratio index (APRI) [29,30], and fibrosis index based on the four factors (FIB-4 index) [31]. Model for end-stage liver disease score (MELD) [32,33], albumin‒bilirubin (ALBI) score [34] and Child‒Pugh score have been associated with postoperative complications [35]. Type IV collagen and serum hyaluronic acid predict post-hepatectomy liver failure and correlate with HF stage [36,37,38,39]. However, the combined evaluation of these images and blood tests has not been adequately investigated. Thus, the combination of images and blood tests for HF may be useful for estimating HF in preoperative patients. Further analysis is recommended.

Ichikawa et al. reported that D* was 0.904 for severe HF (≥F3–4) and 0.885 for liver cirrhosis (≥F4), whereas MR elastography showed D*values of 0.995 for severe HF (≥F3–4) and 0.996 for liver cirrhosis (≥F4). The correlation coefficient of f was greater than that of D* because f correlated best with the HF stage in our study. The results of the HF estimation of f in IVIM were 0.680 and 0.683 for severe HF (≥F3–4), 0.620 and 0.575 for liver cirrhosis (≥F4). The reason for the different results is unknown, but fibrosis assessment by liver biopsy and hepatic resection was 38 and 91, respectively. Unreliable pathological assessment of HF by liver biopsy may be responsible for this. However, our staging of HF was based on the pathology of resected liver specimens, and we believe that the staging was more reliable. In addition, our results showed that each IVIM parameter had non-negligible differences in measurements between the two observers. The lack of stable results in quantitative evaluations remains a cause for concern.

Our study had several limitations. First, it included a small number of patients. Further studies with larger numbers of patients are recommended to confirm our results. Second, the IVIM parameters were acquired using three MR systems. Thus, the results may be influenced by the differences in the MR equipment.

In conclusion, the diagnostic performance of CTV was superior to that of IVIM in patients with severe HF, and the agreement rate of IVIM measurements was not high. We believe that measuring the SV/BSA and estimating the degree of HF in candidates for hepatic resection can contribute to safe and low-risk complications.

## Figures and Tables

**Figure 1 diagnostics-13-03200-f001:**
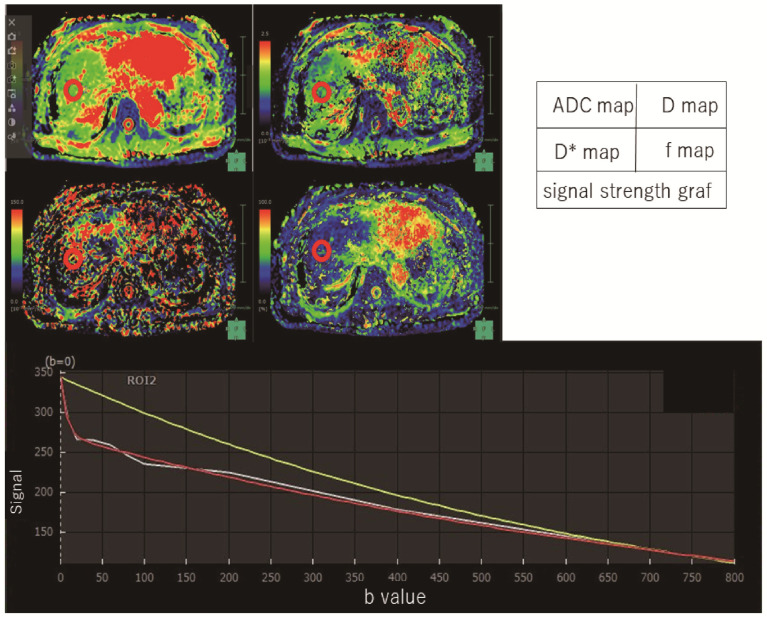
Intravoxel incoherent motion measurement. By measuring three regions of the liver (anterior and posterior segments of the right lobe and lateral segment of the left lobe), IVIM parameters were measured in this patient (F1). This case showed images with an ROI placed in the anterior region of the right lobe of the liver. A model of signal strength variation in IVIM analysis was made from Synapse Vincent software. White line is actual data obtained. Red line is the IVIM nonlinear regression fit providing slow diffusion-coefficient related to molecular diffusion (true diffusion-coefficient) as D, fast diffusion-coefficient related to perfusion in micro-vessels (pseudodiffusion-coefficient) as D*. Yellow line is the monoexponential fit providing apparent diffusion-coefficient (ADC).

**Figure 2 diagnostics-13-03200-f002:**
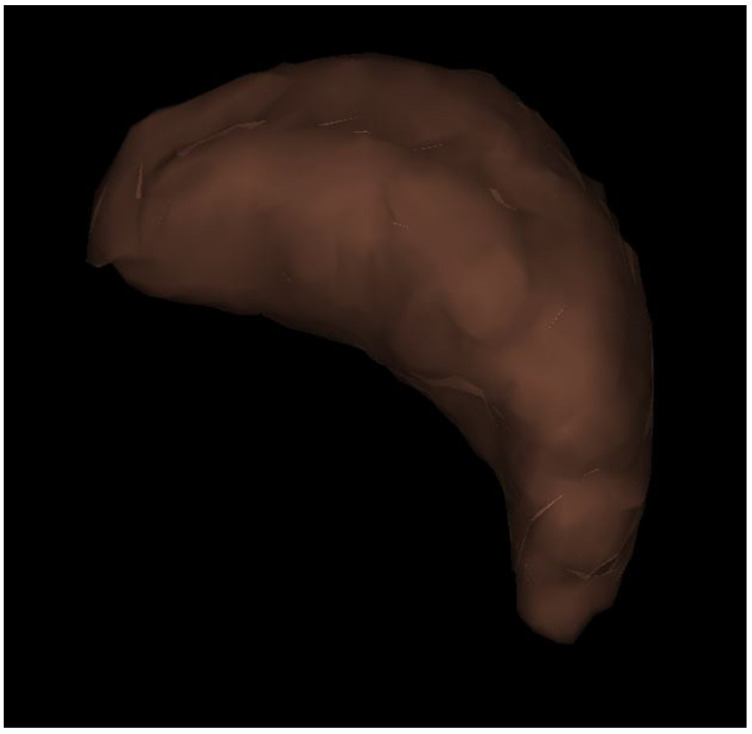
Three-dimensional reconstruction image of the spleen. Splenic hilum view; The volume of the spleen was 74 mL in this patient (F1).

**Figure 3 diagnostics-13-03200-f003:**
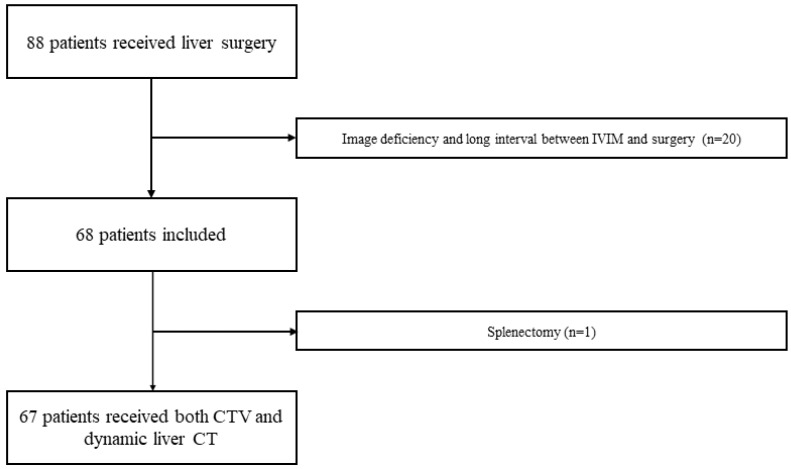
Patient flowchart. A total of 88 patients with liver surgery received both IVIM of MRI and liver dynamic CT. Twenty patients with image deficiency and long intervals between IVIM and surgery were excluded. One of the remaining 68 patients was excluded from the study because he had a splenectomy. Finally, 67 patients were included in the study.

**Figure 4 diagnostics-13-03200-f004:**
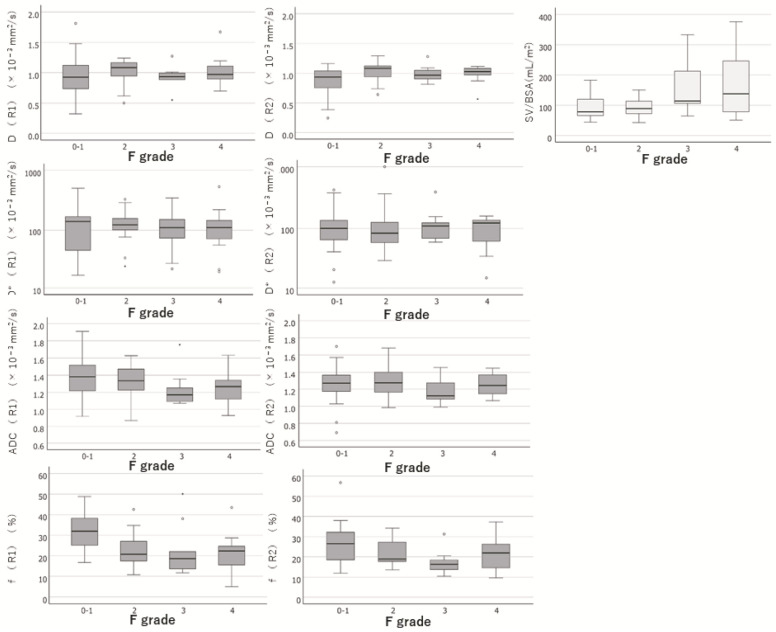
Box-and-whisker diagrams for each parameter measurement. Box-and-whisker diagrams for each parameter measurement of IVIM and SV/BSA were shown. R1, Observer 1; R2, Observer 2; D, molecular diffusion; D*, fast diffusion-coefficient related to perfusion in microvessels; ADC, apparent diffusion-coefficient; f, perfusion-related diffusion fraction; SV/BSA, ratio of splenic volume to body surface area; F grade, hepatic fibrotic grade.

**Table 1 diagnostics-13-03200-t001:** Imaging parameters.

Parameter	T2 Weighted Image	Diffusion Weighted Image	Contrast Enhanced Image
	Discovery/Ingenia/Achieva	Discovery/Ingenia/Achieva	Discovery/Ingenia/Achieva
Sequence	FSE/FSE/FSE	S-E E-P/S-E E-P/S-E E-P	3D GE/3D GE/3D GE
Fat supression	Yes/Yes/Yes	Yes/Yes/Yes	Yes/Yes/Yes
Respiratory triggered	No/No/No	Yes/Yes/Yes	No/No/No
Repetition time (msec)	3500/2100/3200	Variable/1600–2000/1500–2500	4.8/3.1/4.3
Echo time (msec)	100/80/90	65/73/72	2.1/1.48/2.1
Flip angle (degrees)	111/90/90	90/90/90	12/12/12
Parallel imaging factor	sense 2.5/CS2.7/sense 2.4	ASSET 2/sense 2.0/sense 2.0	ARC factor 2.0 2.0/CS3.0/sense 1.8
Field of view (cm)	38/38/38	37/38/38	38/38/38
Matrix	320 × 224/295 × 225/260 × 157	96 × 160/152 × 129/140 × 85	320 × 192/252 × 178/292 × 193
Section thickness (mm)	7/7/7	7/6/6	4.4/4/5
Intersection gap (mm)	1/1/1	1/1/1	0/0/0
Acquisition time	18 s/18 s × 2/19 s	3 min/1 min 50 s/1 min 30 s	14 s/15 s/18 s
b factor		0, 1000/0, 1000/0, 1000	

Note Discovery, Discovery 750 3.0T; Ingenia, Ingenia Elition 3.0T; Achieva, Achieva 1.5T Nova, FSE, Fast spin-echo; S-E E-P, Spin-echo echo-planar; 3D GE, 3-Dimentional gradient echo; CS, compressed sense; ASSET, Array Spatial Sensitivity Encoding Techniques; ARC, Autocalibrating Reconstruction for Cartesian imaging.

**Table 2 diagnostics-13-03200-t002:** IVIM parameters.

	TR/TE (ms)	FA	Matrix	FOV(cm)	Thickness /Gap (mm)	Number of Slices	PI Factor	Number of Examitations	Aqusition Time
Discovery	9230/71.2	90	128 × 128	48.5 × 32.0	7/1	24	(Asset) 2	1 (b = 0–600), 2 (b = 800)	5 m 30 s
Ingenia	2030/70	90	116 × 92	32.0 × 25.3	6/1	35	(sense) 2	2	(-)
Achieva	2160/70	90	144 × 142	39.0 × 38.4	6/1.4	35	(sense) 2	2	(-)

Discovery, Discovery 750 3.0T; Ingenia, Ingenia Elition 3.0T; Achieva, Achieva 1.5T Nova; TR; repetition time, TE; echo time, FA; flip angle, FOV; field of view, PI; parallel imaging; (-); Aqusition time varies between individuals due to respiratory variability.

**Table 3 diagnostics-13-03200-t003:** Patients characteristics.

Sex, *n* (%)	
male	49 (73.1%)
female	18 (26.9%)
Age (year), mean (SD)	70.3 (8.27)
BMI (kg/m^2^), mean (SD)	23.6 (3.79)
Background liver disease, *n* (%)	
HBV	15 (22.4%)
HCV	17 (25.4%)
Alcoholic liver disease	9 (13.4%)
Unknown	26 (38.8%)
Child-Pugh score, *n* (%)	
5	62 (92.5%)
6	4 (6.00%)
7	1 (1.50%)
Pathological F grades, *n* (%)	
F0	10 (14.9%)
F1	14 (20.9%)
F2	18 (26.9%)
F3	10 (14.9%)
F4	15 (22.4%)
Laboratory data	
Hct (%), mean (SD)	39.7 (4.83)
AST (IU/L), mean (SD)	35.4 (22.9)
ALT (IU/L), mean (SD)	31.7 (25.7)
Plt (10^9^/L), mean (SD)	160 (62.8)
INR, mean (SD)	1.01 (0.06)
T-bil (mg/dL), mean (SD)	0.69 (0.32)
Alb (g/dL), mean (SD)	4.17 (0.40)
Cr (mg/dL), mean (SD)	0.86 (0.28)
ICG-R15 (%), mean (SD)	13.6 (10.3)
ALBI, mean (SD)	−2.86 (0.3)
grade 1 *, *n* (%)	48 (71.6)
grade 2a *, *n* (%)	16 (23.9)
grade 2b *, *n* (%)	3 (4.5)
grade 3 *, *n* (%)	0 (0)
MELD, mean (SD)	3.02 (3.36)
APRI, mean (SD)	0.41 (0.16)
FIB-4, mean (SD)	2.13 (1.35)

Notes: BMI, body mass index; HBV, hepatitis B virus infection; HCV, hepatitis C virus infection; ALBI, albumin-bilirubin grade; MELD, model for end-stage liver disease score; APRI, aspartate aminotransferase-platelet ratio index; FIB-4, fibrosis index based on the four factors; Hct, hematocrit; AST, aspartate aminotransferase; ALT, alanine aminotransferase; Plt, platelet; INR, international normalized ratio; T-Bil, total bilirubin; Alb, albumin; Cr, creatinine; ICG-R15, indocyanine green retention rates at 15 min after injection. *; modified ALBI grade.

**Table 4 diagnostics-13-03200-t004:** IVIM and CT Volumetry for each liver fibrosis stage.

		F0–1 (*n* = 24)	F2 (*n* = 18)	F3 (*n* = 10)			
**IVIM**							
D (×10^−3^ mm^2^/s)	R1	0.93 ± 0.35	1.03 ± 0.20	0.94 ± 0.18	F4 (*n* = 15)	ρ	*p* value
	R2	0.88 ± 0.23	1.01 ± 0.17	0.99 ± 0.13			
D* (×10^−3^ mm^2^/s)	R1	136.79 ± 113.89	136.97 ± 76.66	123.31 ± 92.77	1.01 ± 0.23	0.045	0.718
	R2	118.53 ± 96.42	149.70 ± 225.26	126.52 ± 97.75	1.00 ± 0.14	0.21	0.087
ADC (×10^−3^ mm^2^/s)	R1	1.38 ± 0.23	1.32 ± 0.19	1.23 ± 0.20	131.32 ± 123.58	−0.081	0.513
	R2	1.25 ± 0.21	1.28 ± 0.17	1.18 ± 0.16	101.70 ± 47.65	0.047	0.708
f (%)	R1	31.86 ± 8.78	22.75 ± 7.97	22.32 ± 12.32	1.25 ± 0.19	−0.26	0.034
	R2	26.73 ± 9.81	21.49 ± 6.05	17.02 ± 5.95	1.25 ± 0.13	−0.079	0.524
**CT Volumetry**					21.50 ± 8.82	−0.42	<0.001
SV/BSA (mL/m^2^)	R1	92.63 ± 39.18	91.69 ± 28.85	155.27 ± 95.57	20.80 ± 7.85	−0.328	0.007

Spearman’s correlation analysis was used to assess the correlation between each measurement and fibrosis stage. Data are persented as mean ± standard deviation. ρ, Spearman’s correlation coefficient; IVIM intravoxel incoherent motion; D, molecular diffusion; D*, fast diffusion-coefficient related to perfusion in micro-vessels; ADC, apparent diffusion-coefficient; f, perfusion-related diffusion fraction; SV/BSA, ratio of splenic volume to body surface area; R1, Observer 1; R2, Observer 2.

**Table 5 diagnostics-13-03200-t005:** Intra-class correlation coefficient for IVIM.

Parameter	ICC [95% CI]
D	0.471 [0.262–0.638]
D*	0.099 [−0.145–0.331]
ADC	0.621 [0.428–0.756]
f	0.602 [0.402–0.742]

ICC, intra-class correlation coefficient; IVIM, intravoxel incoherent motion; CI, confidence interval; D, true diffusion-coefficient; D*, diffusion-coefficient for perfusion; ADC, apparent diffusion-coefficient; f, perfusion fraction. D* had a skewed distribution, so the log-transformed value was used to calculate the intra-class correlation coefficient. ICC of <0.50 defined as poor; 0.50–0.74 as moderate; 0.75–0.90 as good; and ≥0.90 as excellent.

**Table 6 diagnostics-13-03200-t006:** Comparison between SV/BSA and IVIM.

Variable	SV/BSA (R1)	IVIM, f (R1)	IVIM, f (R2)
F0–1 vs. F2–4			
Optimal cutoff value	98.26	23.7	22.95
Sensitivity [%]	0.628	0.744	0.744
Specificity [%]	0.625	0.792	0.625
AUC (95%CI)	0.628 (0.489–0.767)	0.790 (0.678–0.902)	0.705 (0.571–0.840)
*p* value (SV/BSA vs. f)	(-)	0.0968	0.445
F0–2 vs. F3–4			
Optimal cutoff value	189.87	25.6	15.45
Sensitivity [%]	0.36	0.8	0.44
Specificity [%]	1	0.548	0.905
AUC (95%CI)	0.698 (0.556–0.841)	0.680 (0.542–0.818)	0.683 (0.548–0.819)
*p* value (SV/BSA vs. f)	(-)	0.862	0.872
F0–3 vs. F4			
Optimal cutoff value	137.86	28.7	15.45
Sensitivity [%]	0.533	0.933	0.4
Specificity [%]	0.865	0.385	0.827
AUC (95%CI)	0.649 (0.453–0.845)	0.620 (0.460–0.779)	0.575 (0.400–0.750)
*p* value (SV/BSA vs. f)	(-)	0.862 (0.655–1.000)	0.862 (0.655–1.000)

SV, splenic volume; BSA, body surface area; SV/BSA, ratio of SV to BSA; IVIM, intravoxel incoherent motion; f perfusion-related diffusion fraction; AUC, area under the ROC curve; R1, Observer 1; R2, Observer 2. AUCs are shown along with 95% confidence intervals. The AUC, optimal cutoff value, sensitivity, and specificity of SV/BSA and IVIM (f) for identifying fibrosis stages were calculated. SV/BSA and IVIM (f) were compared using the Delong test, respectively.

## Data Availability

Not applicable.

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
