# Peer review of "Comparison between Intravoxel Incoherent Motion and Splenic Volumetry to Predict Hepatic Fibrosis Staging in Preoperative Patients"

_diagnostics, 2023, doi:10.3390/diagnostics13203200_

Round 1
Reviewer 1 Report
Arakane et al. proposed an orginal work concerning MRI data in hepatic fibrosis. They show a potential interest of IVM and splenic volumetry.
Some elements msut be improved:
- Child Pugh score has no interest in absence of cirrhosis. Furthemore, a patient between 7 and 15 was reported, the real score must be written.
- Etiologies calculation is difficult to be understood, in the same table. A lack of data?
Author Response
Point by point for reviewer 1
Comments and Suggestions for Authors
Arakane et al. proposed an orginal work concerning MRI data in hepatic fibrosis. They show a potential interest of IVM and splenic volumetry.
Some elements must be improved:
- Child Pugh score has no interest in absence of cirrhosis. Furthemore, a patient between 7 and 15 was reported, the real score must be written.
→ One patient with a score between 7 and 15 was listed in Table 2, but the actual score was given as Child-Pugh score was 7.
- Etiologies calculation is difficult to be understood, in the same table. A lack of data?
→ We apologize for any errors in the description.
There were 26 cases with Etiology Unknown, which were described as Others, but we changed it to Etiology unknown.

Reviewer 2 Report
The work is interesting because it delves into the possible role of IVIM-imaging in liver fibrosis assessment. However, the actual clinical benefit compared to numerous non-invasive methods (liver and spleen US elastography, MRI-elastography, laboratory tests) is elusive. - These other methods should at least be named, and if possible thoroughly discussed the reason why IVIM-imaging could provide more information compared to MRI-elastography. -It would be interesting to know IVIM-imaging performance in differentiating F1, F2 and F3 from each other. If there is any data on this please add them.Author Response
Point by point for reviewer 2
Comments and Suggestions for Authors
The work is interesting because it delves into the possible role of IVIM-imaging in liver fibrosis assessment. However, the actual clinical benefit compared to numerous non-invasive methods (liver and spleen US elastography, MRI-elastography, laboratory tests) is elusive.
- These other methods should at least be named, and if possible thoroughly discussed the reason why IVIM-imaging could provide more information compared to MRI-elastography.
→In accordance with the opinion of Reviewer 2, the following text was added to the Discussion regarding imaging and blood tests that can be used as non-invasive methods of hepatic fibrosis (HF) assessment
Non-invasive methods such as US elastography, MRI-elastography, and laboratory tests are used to evaluate HF. Ultrasonography (US) has non-invasive character and FibroScan or Transient Elastography show an excellent ability of liver stiffness, such as the sensitivity of 96.2 %, the specificity of 92.2%, for fibrosis stage ≥4 [26]. MR elastography is useful tool for assessing pathological conditions that affect the elasticity of soft tissues, such as HF [27]. While MR elastography is a non-invasive and provides accurate staging HF [28], its widespread adoption in all hospitals is hindered by the requirement for additional vibration equipment. In recent years, there have been efforts to estimate the degree of HF using existing imaging and blood tests, such as aspartate aminotransferase-platelet ratio index (APRI) [29; 30], and fibrosis index based on the four factors (FIB-4 index) [31]. Model for end-stage liver disease score (MELD) [32; 33], albumin‒bilirubin (ALBI) score [34] and Child‒Pugh score have been associated with postoperative complications [35]. Type IV collagen and serum hyaluronic acid predict post-hepatectomy liver failure and correlate with HF stage [36-39]. However, the combined evaluation of these images and blood tests has not been adequately investigated. Thus, the combination of imagings and blood tests for HF may be useful for estimating HF in preoperative patients. Further analysis is recommended.
References (added)
26 Hashemi SA, Alavian SM, Gholami-Fesharaki M (2016) Assessment of transient elastography (FibroScan) for diagnosis of fibrosis in non-alcoholic fatty liver disease: A systematic review and meta-analysis. Caspian J Intern Med 7:242-252
27 Loomba R, Wolfson T, Ang B et al (2014) Magnetic resonance elastography predicts advanced fibrosis in patients with nonalcoholic fatty liver disease: a prospective study. Hepatology 60:1920-1928
28 Liu H-L, Zhang B, Zhu H, Wang Q-B (2012) Performance of magnetic resonance elastography and diffusion-weighted imaging for the staging of hepatic fibrosis: A meta-analysis. Hepatology 56:239-247
29 Wai CT, Greenson JK, Fontana RJ et al (2003) A simple noninvasive index can predict both significant fibrosis and cirrhosis in patients with chronic hepatitis C. Hepatology 38:518-526
30 Chou R, Wasson N (2013) Blood tests to diagnose fibrosis or cirrhosis in patients with chronic hepatitis C virus infection: a systematic review. Ann Intern Med 158:807-820
31 Vallet-Pichard A, Mallet V, Nalpas B et al (2007) FIB-4: an inexpensive and accurate marker of fibrosis in HCV infection. comparison with liver biopsy and fibrotest. Hepatology 46:32-36
32 Kamath PS, Kim WR (2007) The model for end-stage liver disease (MELD). Hepatology 45:797-805
33 Cucchetti A, Ercolani G, Vivarelli M et al (2006) Impact of model for end-stage liver disease (MELD) score on prognosis after hepatectomy for hepatocellular carcinoma on cirrhosis. Liver Transplantation 12:966-971
34 Pang Q, Zhou S, Liu S, Liu H, Lu Z (2022) Prognostic role of preoperative albumin-bilirubin score in posthepatectomy liver failure and mortality: a systematic review and meta-analysis. Updates in Surgery 74:821-831
35 Au KP, Chan SC, Chok KS et al (2017) Child-Pugh Parameters and Platelet Count as an Alternative to ICG Test for Assessing Liver Function for Major Hepatectomy. HPB Surg 2017:2948030
36 Ishii M, Itano O, Shinoda M et al (2020) Pre-hepatectomy type IV collagen 7S predicts post-hepatectomy liver failure and recovery. World J Gastroenterol 26:725-739
37 Nanashima A, Yamaguchi H, Shibasaki S et al (2001) Measurement of serum hyaluronic acid level during the perioperative period of liver resection for evaluation of functional liver reserve. J Gastroenterol Hepatol 16:1158-1163
38 Chen Z, Ma Y, Cai J et al (2022) Serum biomarkers for liver fibrosis. Clinica Chimica Acta 537:16-25
39 Mak KM, Mei R (2017) Basement Membrane Type IV Collagen and Laminin: An Overview of Their Biology and Value as Fibrosis Biomarkers of Liver Disease. Anat Rec (Hoboken) 300:1371-1390
- It would be interesting to know IVIM-imaging performance in differentiating F1, F2 and F3 from each other. If there is any data on this please add them.
→ We have no data on the differentiation of F1, F2, and F3 other than what is shown in Table 3, Figure 4, and Table 5. As shown in these results, neither IVIM nor SV/BSA have adequate fibrosis-predicting ability in cases of low liver fibrosis
